# Heparin-Binding Protein (HBP), Neutrophil Gelatinase-Associated Lipocalin (NGAL) and S100 Calcium-Binding Protein B (S100B) Can Confirm Bacterial Meningitis and Inform Adequate Antibiotic Treatment

**DOI:** 10.3390/antibiotics11060824

**Published:** 2022-06-19

**Authors:** Maria Obreja, Egidia Gabriela Miftode, Iulian Stoleriu, Daniela Constantinescu, Andrei Vâță, Daniela Leca, Corina Maria Cianga, Olivia Simona Dorneanu, Mariana Pavel-Tanasa, Petru Cianga

**Affiliations:** 1“St. Parascheva” Infectious Diseases Clinical Hospital, 700116 Iasi, Romania; andrei.vata@umfiasi.ro (A.V.); daniela.leca@umfiasi.ro (D.L.); olivia.dorneanu@umfiasi.ro (O.S.D.); 2Department of Infectious Diseases, Faculty of Medicine, “Grigore T. Popa” University of Medicine and Pharmacy, 700115 Iasi, Romania; d.constantinescu@umfiasi.ro (D.C.); corina.cianga@umfiasi.ro (C.M.C.); mariana.pavel-tanasa@umfiasi.ro (M.P.-T.); petru.cianga@umfiasi.ro (P.C.); 3Department of Mathematics, Faculty of Mathematics, “Alexandru Ioan Cuza“ University, 700259 Iasi, Romania; iulian.stoleriu@uaic.ro; 4Immunology Department, “St. Spiridon” Emergency Clinical Hospital, 700111 Iasi, Romania

**Keywords:** bacterial meningitis, biomarkers, HBP, NGAL, S100B, NSE

## Abstract

The empirical administration of antibiotics for suspected bacterial meningitis denotes a poor bacterial stewardship. In this context, the use of biomarkers can distinguish between bacterial and viral infections before deciding treatment. Our study assesses how levels of heparin-binding protein (HBP), neutrophil gelatinase-associated lipocalin (NGAL), S100 calcium-binding protein B (S100B), and neuron-specific enolase (NSE) in cerebrospinal fluid (CSF) and in blood can promptly confirm bacterial etiology and the need for antibiotic treatment. The CSF and blood levels of HBP, NGAL, S100B, and NSE of 81 patients with meningitis were measured and analyzed comparatively. Statistical sensitivity, specificity, and positive and negative predictive values were evaluated. CSF levels of HBP and NGAL and the blood level of S100B in the bacterial meningitis group were significantly higher (*p* < 0.05). The area under curve (AUC) for predicting bacterial meningitis was excellent for the CSF level of HBP (0.808 with 93.54% sensitivity and 80.64% specificity), good for the CSF level of NGAL (0.685 with 75.00% sensitivity and 65.62% specificity), and good for the blood level of S100B (0.652 with 65.90% sensitivity and 57.14% specificity). CSF levels of HBP and NGAL, as well as the blood level of S100B, could help discriminate between bacterial and viral meningitis before considering antibiotic treatment.

## 1. Introduction

Infections of the central nervous system, such as meningitis, are associated with high mortality rates and constitute medical emergencies. In bacterial meningitis, the prompt administration of antibiotics is essential to the survival and recovery of the patient [1]. Since antibiotics are ineffective in viral forms, the differential diagnosis between bacterial and viral meningitis is a priority for making optimal treatment choices [2,3]. However, traditional culturing methods can take several days to render results, and an etiologic bacterial organism is identified in less than one-third of the cases [4]. The incomplete, inaccurate, or delayed diagnosis of viral CNS infections will essentially cancel the intended benefits of antibiotic therapy [3]. In such circumstances, identifying the association of biomarkers in the cerebrospinal fluid (CSF) [5,6,7] and/or blood that best predict the etiology of meningitis (either bacterial or viral) can provide timely clues regarding neuronal damage as well as any bacterial culprits which antimicrobial therapy could neutralize [8].

The heparin-binding protein (HBP), also known as azurocidin, is produced by polymorphonuclear leukocytes under the stimulation of various bacterial agents, cytokines, and inflammatory and chemotactic factors [9,10,11]. Increased levels of HBP have been reported in patients with urinary tract infections, pneumonia, acute respiratory distress syndrome, bacterial skin infections, parasitic infections, bacterial meningitis, infectious endocarditis, and even myocardial infarction [10,12].

Neutrophil gelatinase-associated lipocalin (NGAL) is an essential component of the immune system and plays a role in microbial defense. It is present in the neutrophils which populate various tissues and organs, such as the heart, lungs, liver, kidneys, and brain. NGAL has already been used to differentiate between bacterial-mediated infectious processes and non-bacterial ones. Elevated NGAL has been found in the blood of patients with urinary infection, community-acquired bacterial pneumonia, and sepsis; similarly, patients with bacterial meningitis are known to have high levels of NGAL in their CSF [13,14,15].

S100B is a Ca^2+^-binding protein found mainly in astrocytes. It is produced in higher quantities and released in the CSF, blood, urine, saliva, and amniotic fluid when neurons are being affected. Although it is involved in a wide spectrum of diseases (acute cerebral lesions, neurodegenerative conditions, congenital/perinatal disorders, mental disorders, meningitis), which makes it less specific, measuring it can provide useful clues during patient monitoring [16,17,18].

Neuron-specific enolase (NSE) is a well-established biomarker of neuronal stress and it has prognostic value for a range of neurological disorders. High NSE is a clear indicator of several neurodegenerative conditions, including Friedreich ataxia, hereditary spastic paraplegia, rare forms of Parkinson’s disease, and Alzheimer’s disease. Increased levels of NSE have been significantly correlated with stroke, certain types of cancer, and brain lesions following heart surgery and cerebral infarction [19,20]. Both levels of S100B and of NSE were studied as potentially useful biomarkers in patients with bacterial meningitis, with promising results [21].

Our study aimed to assess the applicability of these biomarkers to the timely diagnosis of bacterial etiology in neuromeningeal infections and appropriate subsequent administration of antibiotic therapy. This is, to our knowledge, the first study investigating the levels of HBP, NGAL, S100B, and NSE in both the blood and the CSF of patients diagnosed with meningitis via conventional means.

The practical value of such research is that it integrates a general attitude of antimicrobial stewardship and a precise assessment of biomarkers to facilitate a prompt and informative diagnostic process [22]. The decision to administer antibiotic treatment in full confidence of its appropriateness is not only life-saving for the individual patient, but it also protects future patients from the antimicrobial resistance effects.

## 2. Materials and Methods

### 2.1. General Data

The study enrolled 81 patients admitted to our Infectious Diseases Hospital during February 2018—November 2020. We included in this study patients over the age of 18 who presented signs and symptoms suggestive for meningitis in accordance with the European Society of Clinical Microbiology and Infectious Diseases (ESCMID) guideline (fever, headache, neck stiffness, and altered mental status) [23] and whose diagnoses were confirmed cytologically and biochemically by lumbar puncture. All the patients who did not fulfill the mentioned criteria or did not consent to the research were excluded.

Our patients (*n* = 81) were further grouped in two main categories: one represented by bacterial meningitis (*n* = 47) and the other one by viral meningitis (*n* = 34). In the bacterial meningitis group, we included the patients who associated CSF modifications, such as pleocytosis of mainly polymorphic leukocytes, low glucose concentration, and elevated protein levels. Identification of bacteria either directly by Gram stain smears and cultures from blood/CSF or indirectly by latex agglutination test of CSF confirmed the bacterial infections. Viral meningitis was defined as the presence of acute onset of meningitis symptoms, WBC of >15/mm^3^ with predominance of mononuclear/lymphocyte cells in the absence of any bacterial meningitis laboratory criteria [23].

### 2.2. Methods

General and clinical patient data were recorded on admission, when blood and CSF were also drawn for laboratory testing and subsequent analysis. Five mL of venous blood and three mL of CSF were centrifuged at 2000 rpm and frozen until the levels of HBP, NGAL, S100B, and NSE could be measured using DuoSet ELISA Development system kits (R&D Systems, Minneapolis, MN, USA, #DY5169-05, #DY1757-05, #DY1820-05) and the DuoSet Ancillary Reagent Kit 2 (R&D Systems, DY008) according to the manufacturer’s instructions. The HBP levels were quantified using Human Azurocidin ELISA kits (Novus Biologicals, Littleton, CO, USA, NBP2-79502).

A Tecan (Austria) microplate reader was used to read the absorbance in the working wells. The standard curves were generated with Magellan software and further used to calculate the concentrations of the investigated biomarkers in the samples. When the detected concentrations were higher than standard, the respective samples were retested after appropriate dilution.

### 2.3. Statistical Analysis

The statistical analysis was conducted using IBM SPSS Statistics for Windows, Version 23.0 (Released 2015, Armonk, NY, USA: IBM Corp.) Mean, median, and standard deviation (SD) values were computed, and the results of the two patient groups were compared using the Mann–Whitney test. Ordinal data were expressed in percentages (%) and compared using the chi2 test. The risk factors were assessed using odds ratio (OR) and 95% confidence interval (CI). For biomarkers accuracy, the receiver operating characteristic (ROC) curve and area under curve (AUC) were constructed, the optimal cut-off value was determined, and the performance parameters such as sensitivity (Se), specificity (Sp), positive predictive value (PPV), and negative predictive value (NPV) were calculated. The threshold for statistical significance was set at *p* < 0.05.

## 3. Results

Of the 81 patients enrolled in the study, 51 were men and 30 were women, with ages raging between 18 and 96 years (54.60 ± 18.69 years). Bacterial meningitis was confirmed in 47 cases, while the other 34 patients had viral meningitis. The mean age in the bacterial meningitis group was lower than in the viral meningitis group (52.72 vs. 57.20). Generally, patients confirmed with bacterial meningitis presented to our hospital sooner after the onset of the symptoms than those diagnosed with viral infection (4.14 vs. 4.73 days); however, they required longer hospitalization (18.74 vs. 15.74 days) (see Table 1).

Patients with bacterial meningitis were mostly men (61.7%) and came from rural areas (61.7%). These and other general characteristics are summarized in Table 2.

Generic symptoms were more common among patients with bacterial meningitis than in the viral group, although the differences were not statistically significant: fever (85.1% vs. 73.5%), headache (91.5% vs. 79.4%) and altered mental status (59.6% vs. 47.1%). The same can be stated about the suggestive manifestations of meningitis, the most common of which were nuchal rigidity (83.0% vs. 82.4%), the Brudzinsky sign (51.1% vs. 41.2%), and the Kernig sign (38.3% vs. 29.4%). Likewise, the computer tomography (CT) scans revealed more lesions in cases of bacterial meningitis (40.4% vs. 35.3%) (see Table 3).

The paraclinical investigations resulted in higher median values for white blood cells (WBC) (15,573 cmm vs. 11,708 cmm), polymorphonuclear leukocytes (PMN) (82% vs. 77%), C-reactive protein (CRP) (122 mg/dL vs. 47 mg/dL), erythrocyte sedimentation rate (ESR) (62.73 mm/h vs. 38.46 mm/h), and fibrinogen (Fib) (5.16 g/L vs. 4.28 g/L) in the group with bacterial meningitis compared to the viral one, and the differences were statistically significant (*p* < 0.05). By contrast, the patients with viral meningitis had significantly higher levels of lymphocytes (Ly) (9.8% vs. 14.07%) and creatinine (0.95 mg/dL vs. 1.09 mg/dL) (see Table 4).

After eliminating data sets without HBP, NGAL, S100B, and NSE values, the study groups comprised 72 cases, of which 41 patients had bacterial meningitis and 31 had viral meningitis. The levels of HBP and NGAL in the patients’ CSF correlated significantly with the bacterial etiology (66.00 ± 134.50 vs. 2.38 ± 5.63 ng/mL, *p* < 0.05 and 11,029.39 ± 12,231.18 vs. 8224.67 ± 13,492.39 pg/mL, *p* < 0.05, respectively). In addition, the blood levels of S100B were significantly higher in the group with bacterial meningitis (166.30 vs. 92.04 ng/mL, *p* < 0.05). For the other biomarkers, there were some noticeable differences between the means of the two study groups, but they were not statistically significant (see Table 5).

Regarding the three biomarkers which yielded statistically significant correlations with bacterial meningitis, the sensitivity, specificity, positive and negative predictive values, and the areas under curve were as follows. At a cut-off value of 2.47 ng/mL, CSF levels of HBP had 93.54% sensitivity and 80.64% specificity, with a positive predictive value (PPV) of 82.84%, a negative predictive value (NPV) of 67.56%, and an area under curve (AUC) of 0.808 (Table 6). For CSF levels of NGAL, at a cut-off of 4.996 pg/mL, the sensitivity was 75%, specificity 65%, PPV 73.17%, NPV 67.74%, and AUC 0.685 (Table 6). For blood levels of S100B, at a cut-off of 36.24 ng/mL, the sensitivity was 65%, specificity 57.14%, PPV 70.72%, NPV 51.61%, and AUC 0.652 (see Table 6 and Table 7).

To further validate that our identified biomarkers in CSF (HBP and NGAL) and blood (S100B) are indeed good predictors for bacterial meningitis, we next conducted a logistic regression analysis for defining various association models. The models that best predicted the bacterial infection included the simultaneous use of biomarkers from both CSF and blood. Since the association of either HBP and NGAL in CSF with S100B in blood (model 1), or only the HBP value in CSF with S100B in blood (model 2) generated similar AUC values (0.904 vs. 0.908), it is clear that the finest predictors of bacterial meningitis among the newly investigated biomarkers would be the CSF levels of HBF and blood levels of S100B. If compared to an association model comprising the significant biomarkers identified in the paraclinical blood tests (listed in Table 4), our proposed models (1 and 2) yielded higher goodness-of-fit, as shown by the AUC values corresponding to the ROC curves (Figure 1).

## 4. Discussion

Multiple studies have so far investigated biomarkers in the blood and CSF as features of inflammation in the subarachnoid space that could help make the distinction between bacterial and viral causes, but with less than conclusive results [22,25,26,27]. The levels of C-reactive protein, procalcitonin, ESR, and fibrinogen, traditionally used to assess the inflammatory syndrome, can help tell the difference to a certain degree, but they are not specific enough to adequately assess neuromeningeal infections. That is why, in our study, we were interested in concomitantly evaluating inflammation markers as well as more specific indicators of neuromeningeal affection such as HBP and NGAL, as expressed in the patients’ blood and CSF.

The general, clinical, and paraclinical characteristics of our study cohort match other patient data from the literature. As elsewhere, patients with bacterial meningitis presented earlier and complained mainly of fever, headache, altered mental status, and neck rigidity [28,29]. The levels of HBP in the CSF were significantly higher among the patients with bacterial meningitis compared to those with viral meningitis (66.00 vs. 2.38 ng/mL, *p* < 0.05). This result is consistent with published studies reporting statistically significant differences [30]. At a cut-off value of 2.47 ng/mL in CSF for HBP, was 93.54% sensitive and 80.64% specific, with a PPV of 82.84% and an NPV of 67.56%. The AUC in our study was 0.808 with a 95% confidence interval of 0.709–0.907. In 2018, Kandil et al., reported an overall accuracy of 100% at a cut-off of 56.7 ng/mL in CSF [30]. Other studies documented for HBP levels in CSF to be 97% sensitive and 95% specific, with predictive values of 93% (positive) and 98% (negative) for a cut-off value of 23 ng/mL [31].

For blood levels of HBP, reported mean values were 192.2 ± 56.6 ng/mL in patients with bacterial meningitis and 3.7 ± 1.9 ng/mL for those with viral meningitis [30]. In our study, the mean HBP was higher in the viral meningitis group (18.88 ± 58.13 vs. 4.86 ± 6.71 ng/mL). Although unexpected, this result could be explained by patients’ comorbidities contributing to their heightened inflammatory status.

With regard to NGAL in CSF, levels were significantly higher in patients with bacterial meningitis compared to those with viral meningitis (11,029 ± 12,231.18 vs. 8224 ± 13,492.39 pg/mL, *p* < 0.05), although the differences of NGAL levels in the patients’ blood were not statistically significant. At a cut-off value of 4996.00 pg/mL in CSF, the sensitivity and specificity were calculated at 75% and 65%, respectively, with PPV 73.17%, NPV 67.74%, and AUC 0.685 (95% CI: 0.553–0.816). In the literature, NGAL in CSF has been discussed as a potentially useful marker for differentiating between bacterial meningitis and other CNS infections. For instance, one team of researchers reported sensitivity and specificity values of 88% and 91%, respectively [32]. Likewise, in another study, the levels of NGAL in CSF were significantly higher in patients with bacterial meningitis compared to those with viral forms (125 pg/mL vs. 2 pg/mL at 81% Se, 93% Sp, PPV of 96% and NPV of 71%, *p* < 0.0001) [33].

For S100B and NSE, other published research comparing levels in patients with bacterial and with aseptic meningitis revealed significantly higher values in cases of bacterial meningitis (S100B: 0.569 ng/mL vs. 0.288 ng/mL; NSE: 12.1 ng/mL vs. 7.358 ng/mL) [34]. Our findings point to the same pattern, both in our patients’ CSF (S100B: 1901.99 ng/mL vs. 951.94 ng/mL; NSE: 4674.22 ng/mL vs. 2899.22 ng/mL) as well as blood (S100B: 166.30 ng/mL vs. 92.04 ng/mL, NSE: 4053.48 ng/mL vs. 3154.32 ng/mL), with differences in blood levels reaching statistical significance (*p* < 0.05). Studies into NSE levels are still relatively limited, but promising. For instance, at a cut-off value of 1.21, NSE was found to be 86.7% sensitive and 75.4 specific for tuberculous meningitis [35]. In addition, Gazzollo et al., reported that higher mean S100B in the CSF of patients with bacterial meningitis without encephalitis compared to a control group (1340 vs. 160 ng/mL, *p* < 0.01) [36].

For the purpose of diagnosing bacterial meningitis, Mahalini et al., found it helpful to consider levels of S100B ≥ 54 ng/L in CSF (29% Se, 98% Sp) and ≥177 ng/L in blood (19% Se, 98% Sp) [37]. They reported areas under curve of 0.523 and 0.655, respectively, which were similar to those we were able to calculate (0.591 and 0.652, respectively). By comparison, in our study, at a cut-off value of 540.99 ng/mL, S100B levels in CSF were 63.88% sensitive and 50% specific, while those in the patients’ blood yielded 65.90% sensitivity and 57.14% specificity at a cut-off value of 36.24 ng/mL.

Our study is methodologically relevant in the sense that we were able to enroll a relatively large number of patients and measure multiple biomarkers in both blood and CSF, while confirming etiology via conventional means. Concurrently, it is limited in scope as we did not, at this point, aim to report on other possible correlation analyses, such as with the main inflammation markers and other laboratory test results, especially CSF-related. At this time, we express our interest in investigating further if the biomarkers featured in this study could be used to produce viable diagnostic and prognostic models, and we invite interested peer researchers to join us in promoting the judicious emergency use of antibiotics.

## 5. Conclusions

This is the first study that reports on the expression in both cerebrospinal fluid and blood of multiple biomarkers indicative for neuromeningeal infections. In our assessment, levels of HBP and NGAL in CSF, as well as of S100B in the blood of patients suspect of meningitis can provide fast confirmation of bacterial etiology and inform the decision to administer antibiotic therapy swiftly, before the results of conventional cultures are available. This approach combines precision, speed, and judicious use of antibiotics in the diagnosis and treatment of bacterial meningitis.

## Figures and Tables

**Figure 1 antibiotics-11-00824-f001:**
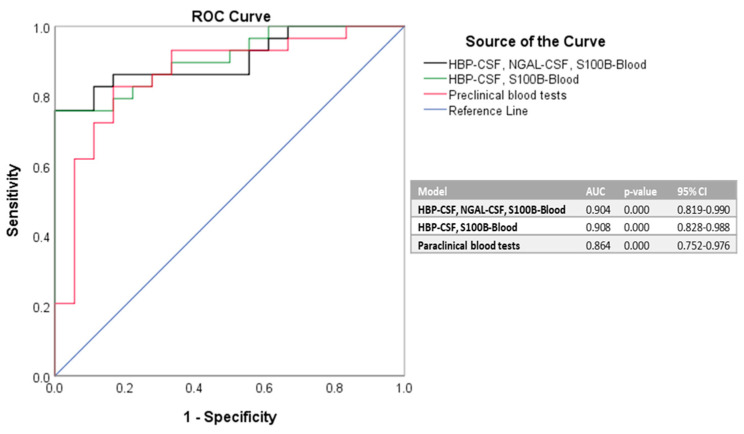
**ROC curves for various associations of biomarkers.** Model 1 comprises the values of HBP and NGAL in CSF and S100B in blood, model 2 comprises the values of HBP in CSF and S100B in blood, while model 3 relies on the significant biomarkers identified during the paraclinical blood tests (WBC, PMN, Ly, CRP, ESR, Fib, creatinine). The AUC values > 0.9 define an outstanding discrimination, while the AUC values between 0.8 and 0.9 denote an excellent capacity for the prediction models [24].

**Table 1 antibiotics-11-00824-t001:** General characteristics of patients.

	Bacterial Meningitis *n* = 47	Viral Meningitis *n* = 34	Total *n* = 81
Age (years), mean ± SD	52.72 ± 20.03	57.20 ± 16.60	54.69 ± 18.69
Onset (days), mean ± SD	4.14 ± 3.41	4.73 ± 5.32	4.39 ± 4.30
Hospitalization (days), mean ± SD	18.74 ± 11.35	15.74 ± 6.13	17.48 ± 9.58

**Table 2 antibiotics-11-00824-t002:** Risk factors for bacterial meningitis.

	Bacterial Meningitis *n* = 47	Viral Meningitis *n* = 34	Total *n* = 81	*p*-Value	OR	95% CI
Sex, Male(%) Female(%)	29(61.7) 18(38.3)	22(64.7) 12(35.3)	51(63.0) 30(37)	0.782	1.13 0.92	0.455–2.846 0.515–1.649
Area, Rural(%) Urban(%)	29(61.7) 18(38.3)	17(50.0) 17(50.0)	46(56.8) 35(43.2)	0.294	1.61 1.30	0.659–3.936 0.796–2.141
Risk factors Alcoholism(%) Smoking(%)	21(44.7) 7(14.9)	12(35.3) 9(26.5)	33(40.7) 16(19.8)	0.396 0.197	1.48 0.48	0.597–3.673 0.161–1.471

**Table 3 antibiotics-11-00824-t003:** Clinical signs and symptoms.

Clinical Signs and Symptoms	Bacterial Meningitis *n* = 47	Viral Meningitis *n* = 34	Total *n* = 81	*p*-Value	OR	95% CI
Altered mental status (%)	28(59.6)	16(47.1)	44(54.3)	0.264	1.65	0.681–4.039
Fever (%)	40(85.1)	25(73.5)	65(80.2)	0.197	2.05	0.680–6.223
Headache (%)	43(91.5)	27(79.4)	70(86.4)	0.117	2.78	0.745–10.427
Vomiting (%)	8(17.0)	7(20.6)	15(18.5)	0.683	0.79	0.256–2.442
Photophobia (%)	2(4.3)	6(11.8)	8(7.4)	0.203	0.33	0.057–1.937
Nuchal rigidity (%)	39(83.0)	28(82.4)	67(82.7)	0.941	1.04	0.326–3.347
Brudzinsky sign (%)	24(51.1)	14(41.2)	38(46.9)	0.379	1.49	0.612–3.633
Kernig sign (%)	18(38.3)	10(29.4)	28(34.6)	0.407	1.49	0.580–3.827
CT (%)	29(61.7)	23(67.6)	52(64.2)			
CT lesions (%)	19(40.4)	12(35.3)	31(38.3)	0.639	1.24	0.499–3.101

CT, computer tomography; OR, odds ratio.

**Table 4 antibiotics-11-00824-t004:** Paraclinical blood test results.

Paraclinical Blood Test Results	Bacterial Meningitis *n* = 47	Viral Meningitis *n* = 34	Total *n* = 81	*p*-Value Mann–Whitney Test
WBC (cmm)	15,573 ± 7561	11,708 ± 4546	13,951 ± 6714	0.020
PMN (%)	82.92 ± 10.21	77.62 ± 11.87	80.70 ± 11.18	0.034
Ly (%)	9.8 ± 8.1	14.07 ± 8.7	11.60 ± 8.6	0.011
CRP (mg/dL)	122.29 ± 104.25	47.62 ± 91.97	91.92 ± 105.33	0.000
ESR (mm/h)	62.73 ± 41.33	38.46 ± 34.18	52.62 ± 40.11	0.007
Fib (g/L)	5.16 ± 2.46	4.28 ± 4.40	4.82 ± 3.33	0.017
Hb (g/dL)	12.68 ± 2.14	12.52 ± 2.34	12.62 ± 2.21	0.837
PLT (mcL)	229,172 ± 141,070	258,411 ± 123,505	241,445 ± 133,962	0.139
Urea (mg/dL)	46.69 ± 42.52	51.55 ± 29.11	48.73 ± 37.35	0.155
Creatinine (mg/dL)	0.95 ± 0.46	1.09 ± 0.41	1.01 ± 0.44	0.046
Glucosemg/dL)	183.52 ± 118.02	146.75 ± 6819	168.16 ± 101.37	0.122
ALT (U/L)	58.80 ± 46.38	143.95 ± 543.40	94.55 ± 353.31	0.358
AST (U/L)	67.69 ± 79.27	117.08 ± 398.73	88.68 ± 265.69	0.138
Bil (mg/dL)	1.19 ± 0.86	0.95 ± 0.70	1.09 ± 0.80	0.061

WBC, white blood cells; PMN, polymorphonuclear cells; Ly, lymphocyte cells; CRP, C-reactive protein; ESR, erythrocyte sedimentation rate; Fib, fibrinogen; Hb, hemoglobin; PLT, platelets cells; ALT, alanine aminotransferase; AST, aspartate aminotransferase; Bil, bilirubin.

**Table 5 antibiotics-11-00824-t005:** HBP, NGAL, S100B, NSE values in CSF and blood.

Studied Biomarkers	Bacterial Meningitis *n* = 41	Viral Meningitis *n* = 31	Total *n* = 72	*p*-Value Mann–Whitney Test
HBP in CSF (ng/mL)	66.00 ± 134.50	2.38 ± 5.63	39.90 ± 107.59	0.000
NGAL in CSF (pg/mL)	11,029.39 ± 12,231.18	8224.67 ± 13,492.39	9822.30 ± 12,781.35	0.008
S100B in CSF (ng/mL)	1901.99 ± 3663.14	951.94 ± 1549.94	1493.11 ± 2968.15	0.189
NSE in CSF (ng/mL)	4674.22 ± 6589.24	2899.22 ± 3034.87	3910.29 ± 5401.04	0.108
HBP in blood (ng/mL)	4.86 ± 6.71	18.88 ± 58.13	10.87 ± 38.69	0.342
NGAL in blood (pg/mL)	5888.67 ± 5816.67	12,532.63 ± 15,461.81	8736.08 ± 11,434.83	0.303
S100B in blood (ng/mL)	166.30 ± 210.05	92.04 ± 130.15	134.48 ± 182.93	0.028
NSE in blood (ng/mL)	4053.48 ± 2843.24	3145.32 ± 2031.94	3664.27 ± 2552.83	0.189

CSF, cerebrospinal fluid; HBP, heparin-binding protein; NGAL, neutrophil gelatinase-associated lipocalin; S100B, calcium-binding protein B; NSE, neuron-specific enolase.

**Table 6 antibiotics-11-00824-t006:** AUC for all biomarkers (HBP, NGAL, S100B, NSE) in CSF and blood.

Studied Biomarkers	AUC	SE	*p*-Value	95%CI
HBP in CSF	0.808	0.50	0.000	0.709–0.907
NGAL in CSF	0.685	0.67	0.008	0.553–0.816
S100B in CSF	0.591	0.68	0.189	0.457–0.725
NSE in CSF	0.611	0.68	0.108	0.478–0.745
HBP in blood	0.434	0.69	0.342	0.299–0.569
NGAL in blood	0.409	0.71	0.189	0.270–0.548
S100B in blood	0.652	0.66	0.028	0.523–0.780
NSE in blood	0.597	0.68	0.160	0.465–0.730

CSF, cerebrospinal fluid; HBP, heparin-binding protein; NGAL, neutrophil gelatinase-associated lipocalin; S100B, S100 calcium-binding protein B; NSE, neuron-specific enolase; AUC, area under curve; SE, standard error.

**Table 7 antibiotics-11-00824-t007:** Statistical evaluation of biomarkers (HBP, NGAL, S100B, NSE) in CSF and blood.

Studied Biomarkers	Cut-Off	Se	Sp	PPV	NPV
HBP in CSF (ng/mL)	2.47	93.54	80.64	82.84	67.56
NGAL in CSF (ng/mL)	4996.00	75.00	65.62	73.17	67.74
S100B in CSF (ng/mL)	540.99	63.88	50.00	56.09	58.06
NSE in CSF (ng/mL)	2216.25	66.66	56.25	63.41	58.06
HBP in blood (ng/mL)	2.37	55.26	41.17	52.50	45.16
NGAL in blood (pg/mL)	3368.00	53.84	48.07	34.14	80.64
S100B in blood (ng/mL)	36.24	65.90	57.14	70.73	51.61
NSE in blood (ng/mL)	4984.92	56.09	67.74	69.69	53.84

CSF, cerebrospinal fluid; HBP, heparin-binding protein; NGAL, neutrophil gelatinase-associated lipocalin; S100B, S100 calcium-binding protein B; NSE, neuron-specific enolase; Se, sensitivity; Sp, specificity; PPV, positive predictive value; NPV, negative predictive value.

## Data Availability

The data presented in this study are available on request from the corresponding author.

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
