# Peer review of "Heparin-Binding Protein (HBP), Neutrophil Gelatinase-Associated Lipocalin (NGAL) and S100 Calcium-Binding Protein B (S100B) Can Confirm Bacterial Meningitis and Inform Adequate Antibiotic Treatment"

_antibiotics, 2022, doi:10.3390/antibiotics11060824_

Round 1
Reviewer 1 Report
See the attached pdf and use Adobe Acrobat Reader to see my comments

Reviewer 2 Report
The manuscript "HBP, NGAL and s100B can confirm bacterial meningitis and inform adequate antibiotic treatment" is interesting, easy to read and provides useful information for the clinicians. The study aimed to to assess the applicability of biomarkers to the timely diagnosis of bacterial neuromeningeal infections and subsequent administration of antibiotic therapy. Unfortunately, the manuscript has major critical issues that must be overcome before the manuscript can be accepted for publication.
1) The novelty of the study is not high, and the data are mostly not clear and convincing because the biomarkers analyzed by the authors have long been known as biomarkers indicative of neuromeningeal infections (Kandil, M et al 2018; Kong et al 2022). The difference pointed out by the authors with previous work is speculative rather than substantive
2) To produce diagnostic and prognostic models the authors should analyze in more detail the data available to them (representing a significant number of patients) the authors should analyze in more detail the data available to them (representing a large number of patients) by providing the correlation analysis data (e.g., identification of bacteria, how many Gram+/Gram-, how many MDR, etc.)
Reviewer 3 Report
Dear Authors,
This study is important, and once clear biomarkers are found to differentiate meningitis agents quickly and without invasive techniques, precision medicine will go a step further.
However, adjustments need to be made to the presentation of the results and conclusions of your manuscript:
1. All tables require headings that clearly reflect the contents of the tables.
2. Line 126 - Tabel2 correct - Table 2.
3. The content of Table 2 needs to be recalculated because it contains a lot of% errors. Especially the rows that reflect the information about Male and Rural. Also in the column - Total, there are both numeric and% errors.
When correcting the numerical and% errors in Table 2 accordingly, it is necessary to make sure that there are no errors in the text that reflects the information in Table 2.
4. There is no reference to Table 6 anywhere in the text. A reference in the text is needed, or consider the need for this table.
5. Line 184 - [[28]...please correct it.
6. In Conclusions
In both the introduction and the aim of the study, you emphasize the importance of finding biomarkers in the blood to reduce the need for lumbar punctures. However, the results show that more diagnostically important biomarkers in your study are in the cerebrospinal fluid than in the blood.
So conclusion...This approach combines precision, speed, non-invasiveness, and...This section of the conclusions needs to be modified as it has not been possible to prove the presence of biomarkers in the blood that could replace CFS.
Round 2
Reviewer 1 Report
Thank you for the resumbmitted version of your interesting article.
Reviewer 2 Report
The authors have addressed all my comments and made the necessary changes to the manuscript